# Naturalness Assessment of Forest Management Scenarios in *Abies balsamea–Betula papyrifera* Forests

**Sylvie Côté** [1,*] , **Louis Bélanger** [1] , **Robert Beauregard** [1] , **Évelyne Thiffault** [1] and **Manuele Margni** [2]

1   Department of Wood and Forestry Sciences, Université Laval, Quebec City, QC G1V 0A6, Canada; louis.belanger.1@ulaval.ca (L.B.); Robert.Beauregard@sbf.ulaval.ca (R.B.); Evelyne.Thiffault@sbf.ulaval.ca (É.T.)
2   CIRAIG, Polytechnique Montréal, Department of Mathematical and Industrial Engineering, Montreal, QC H3C 3A7, Canada; manuele.margni@polymtl.ca
*   Correspondence: sylvie.cote.14@ulaval.ca; Tel.: +1-418-424-0422

**Abstract:** *Research Highlights:* This research provides an application of a model assessing the naturalness of the forest ecosystem to demonstrate its capacity to assess either the deterioration or the rehabilitation of the ecosystem through different forest management scenarios. *Background and Objectives:* The model allows the assessment of the quality of ecosystems at the landscape level based on the condition of the forest and the proportion of different forest management practices to precisely characterize a given strategy. The present work aims to: (1) verify the capacity of the Naturalness Assessment Model to perform bi-directional assessments, allowing not only the evaluation of the deterioration of naturalness characteristics, but also its improvement related to enhanced ecological management or restoration strategies; (2) identify forest management strategies prone to improving ecosystem quality; (3) analyze the model's capacity to summarize the effect of different practices along a single alteration gradient. *Materials and Methods:* The Naturalness Assessment Model was adapted to the *Abies balsamea–Betula papyrifera* forest of Quebec (Canada), and a naturalness assessment of two sectors with different historical management strategies was performed. Fictive forest management scenarios were evaluated using different mixes of forestry practices. The sensitivity of the reference data set used for the naturalness assessment has been evaluated by comparing the results using data from old management plans with those based on Quebec's reference state registry. *Results:* The model makes it possible to identify forest management strategies capable of improving ecosystem quality compared to the current situation. The model's most sensitive variables are regeneration process, dead wood, closed forest and cover type. *Conclusions:* In the *Abies balsamea–Betula papyrifera* forest, scenarios with enhanced protection and inclusion of irregular shelterwood cuttings could play an important role in improving ecosystem quality. Conversely, scenarios with short rotation (50 years) could lead to further degradation of the ecosystem quality.

**Keywords:** naturalness; forest management intensity; land use intensity; quality of ecosystems; boreal forest

---

## 1. Introduction

Green building conception relies on quantitative tools to evaluate the environmental impact of different building materials. For wood products, the environmental impacts include the effects of forest management strategies and practices on ecosystem quality.

A conceptual model for naturalness assessment in boreal forests has been recently proposed for the assessment of the impact of wood harvesting on the quality of ecosystems [1]. This model has

been developed from the perspective of being used in life cycle assessment (LCA). LCA, by default, uses biodiversity damage [2,3] to evaluate the quality of ecosystem. However, biodiversity based on species count alone does not reflect the multidimensional character of biodiversity, which includes multiple levels of organization: genetic, species, populations, community and ecosystem [3], and might lead to inappropriate conclusions [4,5]. The model generally used in LCA to establish the relationship between land use and biodiversity (i.e., the species–area relationship (SAR)) presents many limitations, and is not appropriate when the habitat modification does not result in species losses [5]. On the other hand, studies investigating the effects of land use on biodiversity often contrast intensive vs extensive uses [6]. For forestry, such a simplistic approach does not allow for the consideration of the full diversity of practices, each implying a different pressure on the environment [7–9]. To overcome these issues, we developed an alternative approach, based on the naturalness concept [10], which focuses on habitat characteristics, and allows the evaluation of various forest management practices along a single bi-directional alteration gradient [1], i.e., forest ecosystem degradation or restoration, related to given forest management strategies. Generally, models proposed up to now in LCA do not account for a possible improvement of habitat condition related to enhanced ecological management strategies and restoration efforts [2,5], despite the fact that these are seen as crucial actions to enhance ecosystem functioning and halt the decline of biodiversity [11,12].

Many authors have proposed the use of the concepts of naturalness and hemeroby in impact evaluation of land use (such as forestry) on the quality of ecosystems in LCA [13–17]. Naturalness is defined as "*the similarity of a current ecosystem state to its natural state*" [10], whereas hemeroby expresses "*distance to nature*" in landscape ecology [15]. The use of these concepts can provide a management guide that overcomes the challenge of data gaps in biodiversity [1]. However, the use of subjective hemeroby or naturalness classes has been criticized [3]. The model developed here for boreal forest provides a single numerical index, a suitable approach for use in LCA [1].

In order to evaluate forest management scenarios, the assessment should go beyond the gradual transformation related to the progressive implementation of the scenario through time [1], and be placed in the context of its continuous application over the whole productive area.

The aim of this study was to test the bi-directional capacity of the recently proposed model for the assessment of the impact of wood harvesting on the quality of ecosystems [1], to evaluate the performance of distinct enhanced ecological management strategies, including restoration efforts, at the landscape level. For this purpose, we used two adjacent territories located in the boreal eastern *Abies balsamea–Betula papyrifera* ecological bioclimatic domain of Quebec (Canada) with different histories of forest management. The specific objectives of the study were to:

1. Determine the naturalness of different mix of forest management practices to evaluate the bi-directional capacity of the model to assess both ecosystem degradation and restoration;
2. Identify forest management strategies prone to improving ecosystem quality based on a naturalness evaluation;
3. Analyze the model's capacity to summarize the effect of different practices along a single alteration gradient.

## 2. Materials and Methods

The impact on ecosystem quality of forest management scenarios involving a mix of different proportions of forestry practices is evaluated using the Naturalness Assessment Model initially developed for the *Picea mariana*–feathermoss ecological domain of Quebec [1]. This model uses indicators of condition and pressure to calculate a unique index of naturalness resulting from the combination of management strategies including conservation, and different silvicultural treatments (e.g., careful logging, plantation of indigenous species and partial cutting) (see Côté et al., 2019 for the full description of the naturalness assessment method). For the purpose of this study, the model was adapted to the context of the boreal eastern *Abies balsamea–Betula papyrifera* ecological bioclimatic domain of Quebec (see Appendix A for details related to model's adaptation).

## 2.1. Test Area

The territory of the Montmorency Experimental Forest, located north of Quebec City and included in the *Abies balsamea–Betula papyrifera* domain, was used as a test area (see Appendix A for localization and historical information). This experimental research station is divided in two sectors, designated as FM-A and FM-B, according their different histories. FM-A has been subject to continuous small-scale commercial harvest since the mid-sixties, while FM-B has been subject to a second wave of large-scale commercial harvest between 1985 and 2008, before being incorporated into the Montmorency Experimental Forest.

## 2.2. Naturalness Assessment

For the *Abies balsamea–Betula papyrifera* domain, the five naturalness characteristics of the model were evaluated using the same indicators and variables used for the *Picea mariana*–feathermoss domain of the Quebec's boreal forest [1] (Table 1), except for composition where merchant volume proportion of *Picea* spp. was used as a surrogate to obviate the lack of composition data for late successional species (see Appendix A for details). The evaluation is realized in two steps: partial naturalness index for condition indicators (condition_pni: pni in lower case) and naturalness degradation potentials (NDP) are first evaluated. To do so, we use respectively curves, relating measures (percentage of area or volume) to condition_pni, and tables relating percentage of forest area by practices to NDP factors, shown in Appendix A. Then, the partial naturalness for each naturalness characteristic (characteristic_PNI: PNI in capital letters) is calculated using corresponding formula as per Table 1. The final result corresponds to the naturalness index (NI) obtained from the arithmetic mean of the five characteristic_PNI. To ease results interpretation the continuous gradients (partial or global naturalness indexes) can be split in classes of 0.2 (0.0–0.2: very altered; 0.2–0.4: altered; 0.4–0.6: semi-natural; 0.6–0.8: near-natural; 0.8–1: natural).

**Table 1.** Partial naturalness index equations for each naturalness characteristic (characteristic_PNI) (source: [1]).

| Naturalness Characteristic | Characteristic_PNI Equation |
|---|---|
| Landscape context | $Context\_PNI = CF\_pni \times (1 - (ANT\_NDP + Wm\_NDP + W\_CC\_NDP))$ |
| Forest Composition | $Compo\_PNI = ((CT\_pni + LS\_pni)/2) \times (1 - (exo\_NDP + CS\_NDP))$ |
| Structure | $Struc\_PNI = ((OF\_pni + IR\_pni)/2) \times (1 - HS\_NDP)$ |
| Dead wood | $DW\_PNI = 1 - DW\_NDP$ |
| Regeneration process | $RP\_PNI = 1 - RP\_NDP$ |

PNI: partial naturalness index for naturalness characteristics; pni: partial naturalness index for condition indicators; NDP: naturalness degradation potential; CF: closed forests; ANT: anthropization; Wm: modified wetlands; W_CC: humid area in clearcut; CT: cover type; LS: late successional species; exo: exotic species; CS: companion species; OF: old forests; IR: irregular stands; HS: horizontal structure; DW: dead wood; RP: regeneration process.

The model's adaptation to a new region requires to ensure the capture of the main ecological issues recognized for the territory under investigation, and to identify appropriate indicators, as well as variables and data sources for the evaluation. We tested two types of data sources for reference data: local historical studies [18–20] or Quebec's reference state registry [21] (Table A1). The curves used for partial naturalness index (condition_pni) evaluation were calibrated according to the reference data set used (Figures A2 and A3 based on reference data from studies and registry respectively), and the factors used for naturalness degradation potential (NDP) evaluation adapted (Tables A2–A5).

## 2.3. Description of Scenarios

Scenarios correspond to different mixes of practices applied in the context of sustainable timber production. To figure out the result at the landscape level, in the current study, each practice representing a scenario component is applied on a constant basis over a given proportion of the productive area. However, this exercise is highly theoretical, considering that, in reality, each scenario is adjusted

through time to maximize the wood production, rather than having each component being applied on a constant basis. Furthermore, the prevalence of spruce budworm epidemics, which affect balsam fir (*Abies balsamea*) landscapes every 30 years [22], is not taken into account here.

Total area is broken down as follows: total area = water area + terrestrial area; terrestrial area = non-forested area + forested area; forested area = protection area + productive area. Protection is applied as a reduction percentage over the total forested area, and practices related to wood procurement are applied over a given proportion of the productive area. The evaluation must cover the whole landscape; therefore, scenario components must encompass 100% of the productive forested area. The forested area excluded from the productive area corresponding to protection will have a natural evolution, for which historical reference data has been used, except for late successional species. The hypothesis for spruce content (LS) in protected areas was reduced to 4%, based on results from secondary forests [23–25], given that protected areas are generally located in previously exploited areas, where spruce seed-trees have been harvested.

The practices considered for scenarios in the productive area were: careful clearcut logging (CL), which corresponds to the cut with regeneration and soil protection required by law for clearcut operations in Quebec [26], forest plantation (PL) and irregular shelterwood cutting (ISC). As a result of the abundance of natural pre-established balsam fir regeneration in these secondary balsam fir forests [20,23,27], careful logging (CL) is the main regeneration method applied in these forests. Plantations (PL) are generally concentrated on rich sites, where natural resinous regeneration is scarce. The irregular shelterwood silvicultural system (ISC) is compatible with forests types driven by partial stand mortality and gap dynamics such as balsam fir forests, and provides a way to maintain old-growth forest attributes [28]. Two levels of protection were considered: initial protection (ip) and enhanced protection (ep). The initial protection represents 24.4% of the forested area for FM-A and 13.3% for FM-B, and corresponds to current protected areas, along with other areas excluded from forest management, due to the various regulations and constraints (e.g., riparian strips, steep slopes etc.). The enhanced protection (ep) corresponds to the initial protection, plus protected areas projects proposed for the Montmorency Forest [29], and represents 31.7% of the forested area for FM-A and 32.2% for FM-B.

For careful clearcut logging (CL), two different rotation lengths were tested: 50 years (CL50) and 70 years (CL70), the first one corresponding to the business-as-usual in commercial balsam fir forests, and the second one to a practice that favors establishment of natural regeneration [23] and carbon sequestration [30]. Plantation with a rotation length of 60 years (PL60) has been used, based on the rotation used in the last sustainable yield calculation. The irregular shelterwood cutting (ISC) corresponds to a sequence of partial cuts organized in space and time to insure the permanency of the forest cover [28]; this practice was simulated as harvests of 33% of the volume every 30 years. Data used to evaluate practices effects come from secondary forests.

Scenario elaboration began with the evaluation of each of the three components with two variants of CL, over the entire productive area separately, in order to evaluate the maximal theoretical effect of each practice. These scenarios were first evaluated with the current level of protection and then with the enhanced protection level. For the scenarios involving practices mixes, the first assessment considered only the enhanced protection level. Each mix involving CL has been evaluated two times: one with CL50 and one with CL70. Proportion of CL tested varied between 90% and 50% of the productive area (by multiple of 10), between 0 and 40% for PL and between 0 and 50% for ISC, and a test considering 50% PL and 50% ISC has also been included (Table 2).

**Table 2.** Description of management scenarios evaluated.

| | Productive Area Proportions by Scenario Component | | |
|---|---|---|---|
| Scenario# | ISC | PL60 | CL |
| 1 | 0 | 1 | 0 |
| 2 | 0 | 1 | 0 |
| 3 | 0 | 0 | 1 |
| 4 | 0 | 0 | 1 |
| 5 | 0 | 0.1 | 0.9 |
| 6 | 0 | 0.2 | 0.8 |
| 7 | 0 | 0.3 | 0.7 |
| 8 | 0 | 0.4 | 0.6 |
| 9 | 0.1 | 0.1 | 0.8 |
| 10 | 0.2 | 0.1 | 0.7 |
| 11 | 0.1 | 0.2 | 0.7 |
| 12 | 0.1 | 0.3 | 0.6 |
| 13 | 0.2 | 0.2 | 0.6 |
| 14 | 0.3 | 0.1 | 0.6 |
| 15 | 0.1 | 0 | 0.9 |
| 16 | 0.2 | 0 | 0.8 |
| 17 | 0.3 | 0 | 0.7 |
| 18 | 0.4 | 0 | 0.6 |
| 19 | 0.5 | 0 | 0.5 |
| 20 | 0.5 | 0.5 | 0 |
| 21 | 1 | 0 | 0 |
| 22 | 1 | 0 | 0 |

Scenario #: scenario number; ISC: irregular shelterwood cutting; PL60: plantation with 60 years revolution; CL: careful logging.

### 2.4. Hypotheses

The hypotheses used for each scenario component are presented in Table 3.

**Table 3.** Hypotheses used for each scenario component.

| Scenario Component | Cover Type (CT: % Prod Area of Coniferous Cover Type) | Late Successional Species (LS: % Merchantable Volume in *Picea* spp.) | Closed Forests (CF: % Productive Area of Forests > 40 Years Old) | Old Forests (OF: % Productive Area of Forests > 80 Years Old) | Irregular Stands (IR: % Productive Area of Irregular Stands) |
|---|---|---|---|---|---|
| CL50 [1] | | 1 | 20 | 0 | 0 |
| FM-A | 77.49 | | | | |
| FM-B | 79.79 | | | | |
| CL70 | | 3.5 | 42.85 | 0 | 0 |
| FM-A | 81.54 | | | | |
| FM-B | 83.84 | | | | |
| PL60 | | 50 | 33.33 | 0 | 0 |
| FM-A | 92.38 | | | | |
| FM-B | 94.64 | | | | |
| ISC | | 15 | 90 | 90 | 90 |
| FM-A | 78.02 | | | | |
| FM-B | 80.41 | | | | |
| Protection | | | | | |
| FM-A | 79.3 | 4 | 79.9 | 23.7 | 17.8 |
| FM-B | 85.7 | 4 | 76.19 | 57.9 | 40 |

[1] CL50: careful logging in 50 years old stands; CL70: careful logging in 70 years old stands; PL60: plantation with 60 years rotation; ISC: irregular shelterwood cutting; prod_area: productive area.

Age structure related to each scenario under sustainable production has been used to evaluate closed and old forests. For example, for a rotation of 50 years, 40% of the productive area will be

in the 10 years old class, 40% in the 30 years old class and 20% in the 50 years old class, therefore, the corresponding proportion of closed forests will be 20% and 0% for old forests (>60 years old). For ISC, 10% of the area are permanent skid trails. Despite the fact that these trails will be part of the future stand, the proportion of CF, OF and IR were set to a maximum of 90% used as a security factor. The proportion of coniferous cover type for PL60 and ISC is based on eco-forest map data for the corresponding origin code in each territory. For CL50 and CL70, the proportion of coniferous cover type is based on data (number of stems) gathered in FM-A [31]. The resulting proportion has been raised of 2.3% in FM-B, based on the differences observed in inventory data between the two territories for diverse practices, to be coherent with the coniferous aggressiveness in that territory. The proportion of spruce (*Picea* spp.) for each practice was estimated using different studies in the Montmorency Forest [23,32,33]. Clearcuts on wetlands and anthropization levels were set to current values, and kept constant in all scenarios. For pni's evaluation, the proportion of the productive area was adjusted to represent the proportion of forest area for CT, LS, OF and IR, and the proportion of terrestrial area for CF.

*2.5. Sensitivity Analysis*

To identify the most sensitive variables of the current naturalness assessment, a sensitivity analysis was performed by varying pni for condition indicators and NDP by ±10%. The same analysis was also performed for each scenario component, using the 100% scenario with enhanced protection, in order to identify the most sensitive variables related to each component.

As naturalness condition indicators are assessed against historical values considered to be representative of the pre-industrial condition, the sensitivity to the choice of reference data set has been evaluated by comparing results obtained when using two different reference data sets (Table A1). The first assessment was performed using values drawn from different studies and old management plans around the territory of Montmorency Forest [18–20]. A second assessment was performed on scenarios initially assessed using values from the reference state Quebec's registry [21], resulting from simulations of the vegetation dynamics considering only natural disturbance regimes. The scenario ranking resulting from the use of the two reference data sets was statistically compared using the Kendall rank correlation coefficient. The statistical test was performed with "R" 3.6.1 [34].

Multiple assessments of the same set of scenarios was then performed to analyze the effect of different combinations of management parameters, such as the proportion of protected areas, plantation rotation lengths and the proportion of *Picea* spp. in plantations.

## 3. Results

Detailed results of each scenarios evaluation, including those performed for sensitivity analysis, are presented as Supplementary Material where the different assessments (ass) are as follows:

ass_1: initial scenarios assessment with enhanced level of protection (ep);
ass_2: scenarios with enhanced level of protection using registry's reference data set;
ass_3: scenarios with the initial level of protection (initial protection: ip);
ass_4: scenarios with PL50 and enhanced level of protection;
ass_5: scenarios with PL50 and initial level of protection;
ass_6: scenarios with PL50 and enhanced level of protection and 90% of spruce at maturity in plantations;
ass_7: scenarios with enhanced protection and alternate set of NDP factors.

*3.1. Naturalness Index for the Current State of the Forest of the Experimental Forest*

The naturalness assessment results for the current state of the forest in FM-A and FM-B are presented in Figure 1, showing intermediary results for condition indicators (pni), results for each naturalness characteristics (PNI) and the resulting naturalness index (NI), which corresponds to the arithmetic mean of the five Characteristic_PNI. Results show a naturalness index (NI) of 0.5294 for

FM-A and 0.4691 for FM-B. Both territories scored in the semi-natural class (NI: 0.4-0.6), despite their different histories. However, FM-B showed a lower naturalness compared to FM-A, mainly related to a poor structural diversity associated with an important deficit of old forests combined with a lack of irregular stands (Figure 1). The structure is altered in FM-A (PNI_Struc= 0.3332) and very altered in FM-B (PNI_Struc= 0.1931). The alteration of structure in FM-A results from the very low ratio of old forests compared with the historical values; whereas, in FM-B, it is both the low ratio of old forests and the lack of irregular stands.

The landscape context of FM-B is also characterized by an important deficit of closed forests where it reaches the altered level. According to the map information, in FM-A less than 10% of the harvested area was in 50 years old stands, whereas in FM-B this proportion reaches 50%. Precommercial thinning covered 9.2% of the forest area in FM-A compared to 23.4% in FM-B, plantation 6.8% in FM-A and 8.8% in FM-B, and partial cuttings, excluding commercial thinning, 5.5% in FM-A and 1.4% in FM-B.

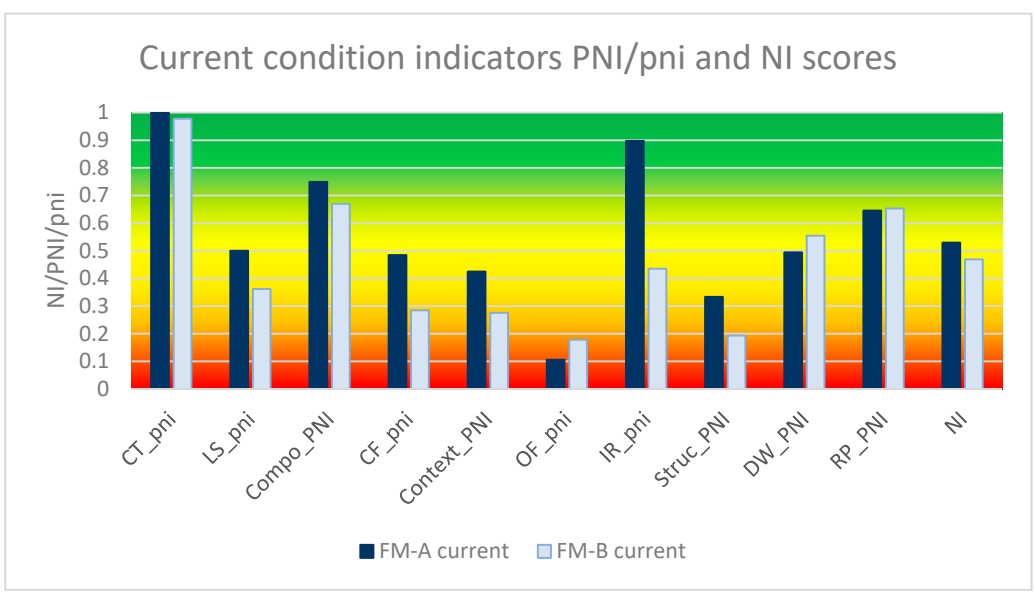

**Figure 1.** Results for condition indicators and naturalness for FM-A and FM-B. PNI: partial naturalness index for naturalness characteristics; pni: partial naturalness index for condition indicators (intermediary results); CT: cover type; LS: *Picea* spp. corresponding to late successional species; Compo: composition; CF: closed forests; Context: landscape context; OF: old forests; IR: irregular stands; Struc: structure; DW: dead wood; RP: regeneration process; NI: naturalness index.

### 3.2. Naturalness of Different Forest Management Scenarios

Scenarios results of the initial assessment are presented in Figure 2 (detailed results in Supplementary Material, file results, page ass_1). Compared with results based on the current state of the forest, some scenarios lead to a lower naturalness index and others to a higher level, indicating the model's capacity to detect not only alteration of the ecosystem quality, but also its improvement.

Scenarios featuring the application of only one practice on 100% (of the productive area) were tested to evaluate their respective extreme theoretical effects on the naturalness assessment resulting from the model. Each of these were evaluated twice: once with the current level of protection (initial protection: ip), and once with the inclusion of the protection projects (enhanced protection: ep). These 100% theoretical scenarios are shown as benchmarks in every test performed to see the maximal theoretical effect related with each scenario component, considering the two levels of protection. Figure 2 shows that enhanced protection impact differs among the scenario components. The positive impact related to protection enhancement is more important when applied concurrently with PL60 than with CL50, CL70 and finally ISC.

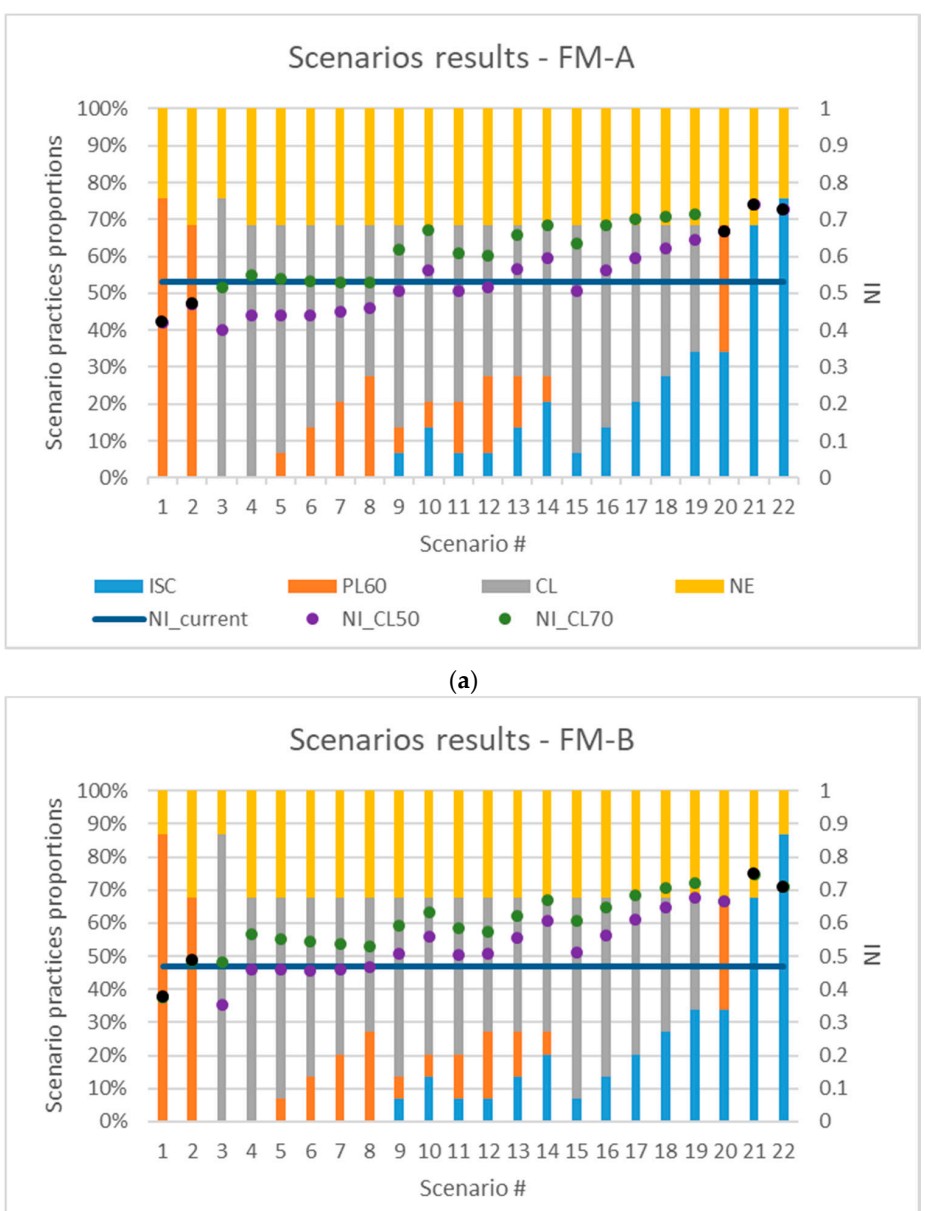

(**a**)

(**b**)

**Figure 2.** Naturalness assessment of forest management scenarios: (**a**) FM-A; (**b**) FM-B. For each scenario, practices mixes are illustrated using the stacked histogram: ISC: Irregular shelterwood cutting; PL60: plantation with 60 years revolution; CL: careful logging; NE: natural evolution (protection); Resulting naturalness index (NI) is indicated with the dots: NI_CL70 (green): naturalness index for CL applied in 70 years old stands; NI_CL50 (purple): naturalness index for CL applied in 50 years old stands; naturalness index for practices other than CL (black); the horizontal solid line shows the current naturalness index (2018) with the initial level of protection.

The PL60 scenario (Sce#1) produces a degradation of the naturalness in both territories. With the inclusion of protection projects (Sce#2), the 100% PL60 scenario still scores lower than current results for FM-A, but slightly better for FM-B. The 100% CL50 scenario is worse than the 100% PL60 scenario, because of the shorter rotation period and a lower amount of spruce. The 100% ISC scenario (Sce#21 and Sce#22) would have the potential to improve naturalness at the quasi-natural level. The quasi-natural level could also be reached in FM-A, with CL70 combined with at least 30% of ISC and enhanced

protection (Figure 2a), and in FM-B with CL70 and enhanced protection with at least 40% ISC (Figure 2b). The 100% CL70 scenario has a NI near to the current level, for the same level of protection. Protection projects represent a greater proportion in FM-B, so the resulting naturalness improvement related to the inclusion of these projects is more important.

The current naturalness in FM-A shows an impact level analog to the 100% CL70 scenario with initial protection, or CL70 with enhanced protection and some plantation. The current level in FM-B presents an impact closer to the 100% CL70 scenario with initial protection, or CL50 with enhanced protection scenarios with some plantation. Considering the intervention mixes, ISC has the potential to compensate at least for a part of the degradation related to forest rejuvenation resulting from careful logging or plantation.

The ISC's potential to improve the overall naturalness is more important when the proportion of ISC is smaller than the historical proportion of irregular stands (FM-A: 17.9%; FM-B: 40%). Above this level, naturalness improvement resulting from a higher level of ISC is less important, as seen in FM-A's CL70 results, where the difference of NI between 40% and 50% of ISC is smaller than the difference between 10% and 20% of ISC. Results suggest that ISC can compensate to some extent for the alteration caused by CL50.

With the hypotheses used, plantations under 60 years rotation, with 50% of the volume in spruce at maturity, could produce an improvement of the naturalness when combined with CL50, mainly related to rotation length.

### 3.3. Sensitivity Analysis

#### 3.3.1. Test of a Variation of 10% of the Parameter Values

The results of the sensitivity analysis performed on current pni and NDP values for the two territories are provided in Figure 3. Due to the use of non-linear models, a uniform variation of input parameters (10%) can have a non-linear effect on the results, depending on the curve slope around the parameter value [1].

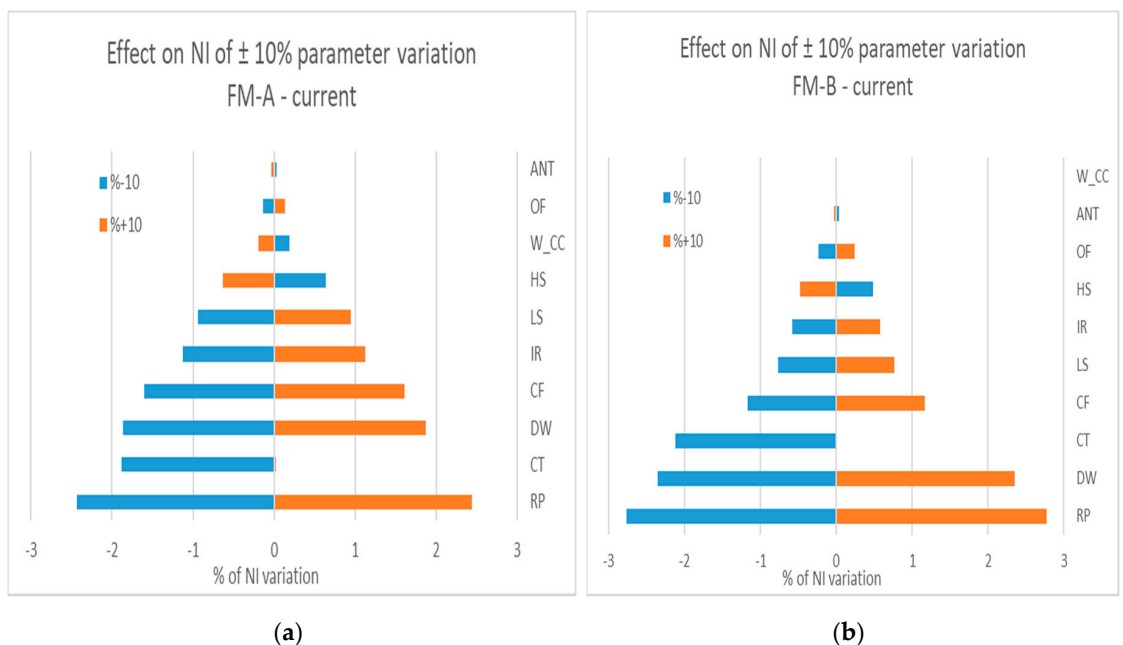

(**a**)                                   (**b**)

**Figure 3.** Results of the sensitivity analysis testing a variation of 10% of the parameter value on the current assessment: CF: closed forests; ANT: anthropization; W_CC: humid area in clearcut; CT: cover type; LS: late successional species; OF: old forests; IR: irregular stands; HS: horizontal structure; DW: dead wood; RP: regeneration process. (**a**) FM-A, (**b**) FM-B.

The most sensitive variables for the naturalness assessment of the current state of the forest are the regeneration process (RP), the cover type (CT), the dead wood (DW) and the closed forest (CF) for both territories; however, dead wood is more sensitive than cover type in FM-B.

The results of the sensitivity analysis performed on the scenarios in which a single component is applied over 100% of the productive area are shown in the Supplementary Material (page sensitivity analysis). As seen from these figures, the most sensitive variables for scenarios involving cuttings (CL, ISC) correspond, in varying order, to the regeneration process, the dead wood, the cover type and the closed forests, whereas the plantation influences mainly closed forest, cover type, late successional companion species and the regeneration process.

### 3.3.2. Test of an Alternative Reference Data Set

Using the registry's reference data set [21], the naturalness of the two territories are closer (FM-A: NIr = 0.4683; FM-B: NIr = 0.4349) (Figure 4) and are both lying in the semi-natural range. The main disparities between the two data sets values are for OF and IR (Table A1) especially for FM-A, leading to a much lower PNI_Struc with the registry's values. The difference related to coniferous cover type between the two data set is not reflected in the model, because of the use of topped curve for CT above the historical value.

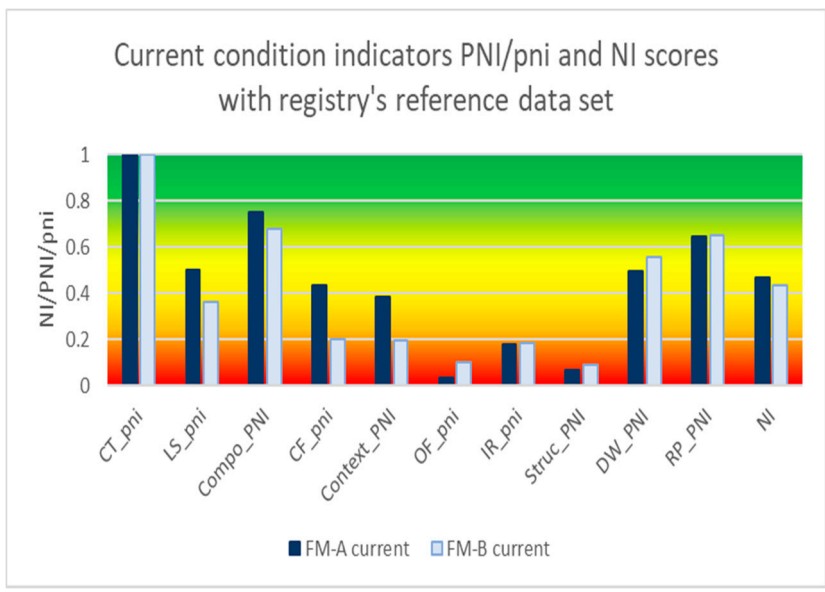

**Figure 4.** Results for condition indicators and naturalness index (NI) for FM-A and FM-B using registry's reference state data set [21].

Scenario results using registry's data as alternative reference data set are presented in Supplementary Material (page ass_2 and Figure S2). The use of the registry's data set places the current assessment for FM-A to a level below CL70 scenarios with some PL60, with initial protection, as well as enhanced protection. We observe the same trends seen with the use of studies as the reference data set: improvement related to use of some ISC, and a slight degradation related to the use of PL60 combined with CL70, but no deterioration (even a slight improvement) with an additional use of PL60 when combined with CL50. However, the reference data set used might affect the identification of the potentially improving or deteriorating scenarios when compared with the current result.

The scenarios were sorted in descending order of NI results using local studies [18–20] as the reference data set versus the registry's reference data [21] (Table 4). The ranking of scenarios is not the same, depending on the reference data set used, but the four best-scoring scenarios are the same (the 100% ISC with both level of protection, and at least 40% ISC combined with CL70 with enhanced protection), as well as the two worst-scoring scenarios (CL50 and PL60 with initial protection).

The remaining 34 scenarios score in the semi-natural class ($0.4 \leq NI < 0.6$). Among these, CL70 with at least 20% ISC and enhanced protection score in the upper part of the class, while CL50 with PL60 and enhanced protection are in the lower part. Among scenario components, the addition of ISC has the biggest positive impact on naturalness, while the addition of PL60 has a negative impact, but is less important when compared to the importance of the positive impact of the ISC.

**Table 4.** Scenario ranking resulting from the use of the two reference data sets for FM-A and FM-B, sorted in descending order of their naturalness index.

| Scenario Sequential nb | Scenario Description [1] | Rank FM-A Studies | Rank FM-A Registry | Rank FM-B Studies | Rank FM-B Registry |
|---|---|---|---|---|---|
| 22 | 100ISC_ep | 1 | 1 | 1 | 1 |
| 23 | 100ISC_ip | 2 | 2 | 3 | 2 |
| 20 | 50CL70_50ISC_ep | 3 | 3 | 2 | 3 |
| 19 | 60CL70_40ISC_ep | 4 | 4 | 4 | 4 |
| 18 | 70CL70_30ISC_ep | 5 | 7 | 5 | 7 |
| 15 | 60CL70_10PL60_30ISC_ep | 6 | 8 | 7 | 8 |
| 17 | 80CL70_20ISC_ep | 7 | 9 | 10 | 10 |
| 11 | 70CL70_10PL60_20ISC_ep | 8 | 11 | 11 | 11 |
| 21 | 50PL60_50ISC_ep | 9 | 5 | 8 | 5 |
| 14 | 60CL70_20PL60_20ISC_ep | 10 | 12 | 12 | 12 |
| 40 | 50CL50_50ISC_ep | 11 | 6 | 6 | 6 |
| 16 | 90CL70_10ISC_ep | 12 | 13 | 14 | 13 |
| 39 | 60CL50_40ISC_ep | 13 | 10 | 9 | 9 |
| 10 | 80CL70_10PL60_10ISC_ep | 14 | 14 | 16 | 16 |
| 12 | 70CL70_20PL60_10ISC_ep | 15 | 15 | 17 | 17 |
| 13 | 60CL70_30PL60_10ISC_ep | 16 | 16 | 18 | 19 |
| 38 | 70CL50_30ISC_ep | 17 | 19 | 13 | 14 |
| 35 | 60CL50_10PL60_30ISC_ep | 18 | 17 | 15 | 15 |
| 34 | 60CL50_20PL60_20ISC_ep | 19 | 25 | 22 | 25 |
| 37 | 80CL50_20ISC_ep | 20 | 26 | 20 | 22 |
| 31 | 70CL50_10PL60_20ISC_ep | 21 | 27 | 21 | 24 |
| 5 | 100CL70_ep | 22 | 18 | 19 | 18 |
| 6 | 90CL70_10PL60_ep | 23 | 20 | 23 | 20 |
| 7 | 80CL70_20PL60_ep | 24 | 21 | 24 | 21 |
| 8 | 70CL70_30PL60_ep | 25 | 22 | 25 | 23 |
| 1 | Current_ip | 26 | 28 | 33 | 33 |
| 9 | 60CL70_40PL60_ep | 27 | 23 | 26 | 26 |
| 4 | 100CL70_ip | 28 | 24 | 32 | 31 |
| 33 | 60CL50_30PL60_10ISC_ep | 29 | 29 | 28 | 28 |
| 32 | 70CL50_20PL60_10ISC_ep | 30 | 31 | 30 | 30 |
| 36 | 90CL50_10PL60_ep | 31 | 32 | 27 | 27 |
| 30 | 80CL50_10PL60_10ISC_ep | 32 | 33 | 29 | 29 |
| 3 | 100PL60_ep | 33 | 30 | 31 | 32 |
| 29 | 60CL50_40PL60_ep | 34 | 34 | 34 | 34 |
| 28 | 70CL50_30PL60_ep | 35 | 35 | 36 | 36 |
| 27 | 80CL50_20PL60_ep | 36 | 36 | 38 | 38 |
| 25 | 100CL50_ep | 37 | 37 | 35 | 35 |
| 26 | 90CL50_10PL60_ep | 38 | 38 | 37 | 37 |
| 2 | 100PL60_ip | 39 | 39 | 39 | 39 |
| 24 | 100CL50_ip | 40 | 40 | 40 | 40 |

[1] ISC: Irregular shelterwood cutting; PL60: plantation with 60 years revolution; CL50: careful logging applied in 50 years old stands; CL70: careful logging applied in 70 years old stands; ip: initial protection; ep: enhanced protection; Current: current naturalness (2018). The digits preceding component code correspond to the percentage of productive area of the component application (ex:100PL60_ep: plantation with 60 years revolution applied over 100% of the productive area with enhanced level of protection).

The use of registry reference data set induces a negative bias compared with the use of the studies reference data set (FM-A: $-0.0591$; FM-B: $-0.0522$), as the naturalness index is generally lower when

registry's data are used. This bias is not constant as a result of the use of nonlinear relationships in the model. Nevertheless, the general trends related to the positive effects of ISC and the relatively limited negative effects of mechanically released plantations, including 50% of spruce at maturity, are observed in the ranking arising from both data sets. The ordinal association between the two rankings, as measured with Kendall rank correlation computed with the base cor function in R, showed a coefficient of 90.5% for FM-A and 95.9% in FM-B. Therefore, the model's aptitude at classifying different scenarios along a single alteration gradient is robust, in regard to the reference data set used. However, assessing improvement or degradation against the current value might be affected by the reference data set used.

### 3.3.3. Naturalness of the Forest Management Scenarios Tested Using Various Hypothesis

Scenarios with Initial Protection

In order to see the impact related to the protection level, scenario evaluation using studies as reference data set was calculated using the current level of protection (ip) (Supplementary Material: page ass_3 and Figure S3). With 24% of the forested area in protection in FM-A, none of the scenarios, except 100% ISC (sce#22), would reach the quasi-natural level, and the scenarios considering CL50 with some PL60 would score around the limit between the altered and the semi-natural levels. The current alteration level is similar to 100% CL70, and scenarios combining with some PL60 are below to the current level even when combined with CL70. In FM-B, with 13% of the forested area in protection, the scenarios considering CL50 with some PL60 would score in the altered level, except when combined with at least 20% of ISC or, at least 10% ISC, if considering CL50 without plantation. The current alteration level in FM-B is similar to 100% CL70, or CL70 with less than 20% PL60, or CL70 with up to 30% PL60, if combined with 10% ISC, with initial protection.

Scenarios with Enhanced Protection and Plantation with 50 Years Rotation

In order to see the impact related to plantation rotation length, an evaluation using studies as the reference data set was calculated using a rotation length of 50 years for plantations (PL50). This test was performed by adjusting the age structure for the planted proportion of the scenario, but keeping all others factors constant (Supplementary Material: page ass_4 and Figure S4).

The 100% PL50 with initial protection scenario would lead to a naturalness index lying in the altered class for both territories (Sce #1: FM-A: NI = 0.3871; FM-B: NI = 0.3308). Adding new protection projects (reaching a total of more than 30% of the forested area) makes it possible to attain the semi-natural class even with 100% PL50 in the productive area (Sce#2: FM-A: NI = 0.4400; FM-B: NI = 0.4511). The use of PL50 instead of PL60 reduces slightly the improvement related to the increasing use of PL, when combined with CL50. We can still observe no further degradation and even a small improvement for Scenario 8 with CL50, compared with Scenario 7 with CL50, related to the increase of spruce species at the landscape level. However, more spruce in the plantations could induce a degradation, as these species were subdominant at the landscape level.

Scenarios with Initial Protection and Plantation with 50 Years Rotation

In order to see the combined impact related to the protection level and plantation rotation, scenario evaluation using studies as reference data set was calculated using the current level of protection and plantation with 50 years rotation (PL50) (Supplementary Material: page ass_5 and Figure S5). Compared with the test considering initial protection only, the use of PL50 instead of PL60 has a limited effect, but pushes more scenarios closer to the altered class in FM-B. With the initial protection and the use of PL50, some ISC is necessary to reach a level above the current naturalness.

Plantation with Enhanced Protection and 50 Years Rotation and 90% of Merchantable Volume in Spruce at Maturity

A test was performed using a proportion of spruce of 90% (instead of 50%) of the merchantable volume at maturity for PL50, in order to analyze the effect of more monospecific plantations with a short rotation (Supplementary Material: page ass_6 and Figure S6). The 100% PL50 with 90% of spruce at maturity scenarios, with initial protection, would lead to a naturalness index lying in the altered class in both territories. The addition of protection projects (reaching more than 30% of the forested area) makes it possible to attain the semi-natural class in FM-B, even with 100% PL50 in the productive area, that would produce 90% of the volume in spruce (Sce#2: FM-A: NI = 0.3998; FM-B: NI = 0.4242). An increasing use of PL50 producing 90% of the volume in spruce, combined with CL50, could improve the naturalness to some extent, until the total spruce proportion of the scenario reaches its historical level. However, above this level, an increase of spruce proportion could lead to a noticeable deterioration, as shown with the 100% PL50 with 90% of spruce (Sce #1 and 2).

Scenarios with Enhanced Protection and a Variant of NDP Factors

The regeneration process and dead wood were among the most sensitive variables for the assessment of the current forest and for the evaluation of scenario components related to harvest (CL, ISC). In order to see the influence of NDP factors, an alternate set of factors, inducing more difference between CL70 and CL50 and ISC, was tested by lowering the three NDP factors (i.e., HS, DW and RP) for CL50 of 0.1, and the NDP factor for partial cutting of 0.1 for HS and RP. Inducing a more important degradation of RP, DW and HS for CL50, and of HS and RP for partial cuttings, results in a slightly higher NI for the assessment of the current forest (FM-A: 0.5550; FM-B: 0.4756). The difference is marginally more important for FM-A, as this territory has less CL50 than FM-B. As expected, this enhances the performance of CL70 scenarios compared to CL50 scenarios, allowing more scenarios to reach the quasi-natural level. On the other hand, the distinction between scenarios that improve naturalness vs. those that degrade it remains the same (Supplementary Material: page ass_7 and Figure S7).

## 4. Discussion

Scenario evaluation suggests that some combinations of practices could produce an improvement of the naturalness index, compared to the state of the current forest, confirming the model's capacity of performing bi-directional assessment to satisfy the need of assessing restoration efforts.

### 4.1. Naturalness Assessment of FM-A And FM-B's Current Forest

Current results in FM-A and FM-B lead to a naturalness index lying between 0.4 and 0.6, corresponding to the semi-natural class. These results consider the current level of protection (ip: FM-A: 23.8%; FM-B: 12.7% of the terrestrial area or 24.4% and 13.3% of the forested area). The difference induced by the distinct forest management strategies applied over the last 50 years is a priori small; it is, however, not surprising, considering that the forest management strategies in both territories were prioritizing the liquidation of old forests using principally single aged management. The lower naturalness observed in FM-B is mainly related to the low level of closed forests (only 27% of the forest area is over 40 years old in FM-B), and a lack of structural diversity (the low level of old forests in FM-B being worsened by the scarcity of irregular stands). However, the contrast between the two territories is limited by the use of the same hypothesis for IR and LS. As the forest rejuvenation was applied more uniformly and over a short period in FM-B, compared with FM-A, a better characterization of the landscape context, taking into account the size of the stands by age-class, could be necessary to include that issue and improve the comparison.

*4.2. Naturalness Evolution through Time*

Results from the Naturalness Assessment Model can be applied to the conceptual time frame of the evolution of land quality with land use intervention generally applied in LCA [35] (Figure 5), considering that the reference state (Qref) corresponds to the natural state (NI = 1). The reference time considered in this study refers to the pre-industrial state, characterized by the spruce budworm epidemics prevailing every 30–40 years in the balsam fir forests [36]. The first cutting cycle (from $t_0$ to $t_1$) corresponds to the transformation phase causing the initial decrease in ecosystem quality, which is progressive in forestry, as shown with the assessment performed in the three Forest Management Units in the *Picea mariana*–feathermoss domain [1]. For a given management strategy, the subsequent rotations (from $t_1$ to $t_2$) cause further degradation, but this is relatively less important than the impact resulting from the initial transformation [1]. For FM-A and FM-B, as no data was available for $t_1$, the same level of naturalness after the first cutting cycle has been assumed. FM-B shows a current naturalness index lower than FM-A, which can be related to the forest management regime applied up to now. Evolution during the sustainable management phase depends upon the forest management scenario applied, and can be further degraded or improved, depending on the forest management strategy, as schematized. A full rotation is necessary to reach the naturalness level evaluated for a given scenario (prior to that, the evaluation would include a part of the previous forest management strategy). Therefore, in order to compare different scenarios, the scenario evaluation must be performed as if the scenario would have been applied over the entire productive area, covering a whole cutting cycle. The conceptual framework for the evolution of ecosystem quality [35] considers an hypothetical relaxation of the land use (at $t_2$), and a progressive return to a level corresponding to the potential natural vegetation (QPNV). The level of quality after relax (QPNV) depends not only on natural vegetation dynamics, but could also be affected by the permanent impact related to practices applied in the past (for example, the introduction of exotic species, disappeared species, forest drainage, etc.). When the land has been used for an extended period of time ($t_2$>>>$t_1$), the natural reference could not be appropriate, the PNV represents an alternative [37]. Our evaluation uses historic pre-industrial data as reference, data from secondary forest to quantify the impact of the silvicultural treatments, and to adjust data for protection to reflect their localization in previously harvested sectors by lowering the proportion of *Picea* spp. However, scenario assessments do not consider future changes resulting from other causes, such as pollution, natural disturbance regime modification related to climate change or other modification of pressures.

*4.3. Scenario Comparison*

The model is used to express the impact level on the environment along a single alteration gradient. Though the choice of the reference data set used to parametrize the curves for condition_pni evaluation might influence the fine ranking of the forestry management practices, in the studied case, the scenarios that could improve naturalness were identified with a good level of confidence.

Among the practices considered, with the hypotheses used, the enhancement of protection combined with the use of ISC provided the best naturalness index scores. Conversely, practices involving short rotation (50 years) lead to the lowest scores. The scenarios are generally leading to the semi-natural range. Reaching of the quasi-natural level is associated with the use of irregular shelterwood cutting (ISC); a proportion close to 100% would be necessary with the initial level of protection. With an enhanced level of protection, the quasi-natural level could be attained when combining CL70 without plantations with at least 30% ISC in FM-A, and at least 40% ISC in FM-B.

On the opposite side of the alteration gradient, with the current level of protection of 13.3% of the forested area in FM-B, the scenario considering clearcuts with a rotation of 50 years (CL50), with or without plantation, would produce an altered level, indicating a potential loss of species, according to the model [1]. In FM-A, with a current level of protection of 24.4% of the forested area, the CL50 scenario with up to 10% PL60 would produce an altered level; scenarios with CL50 combined with more PL60 would also score close to the altered level. The results show the important role of

protection in mitigating forest management effects, from the perspective of limiting the negative effects on biodiversity.

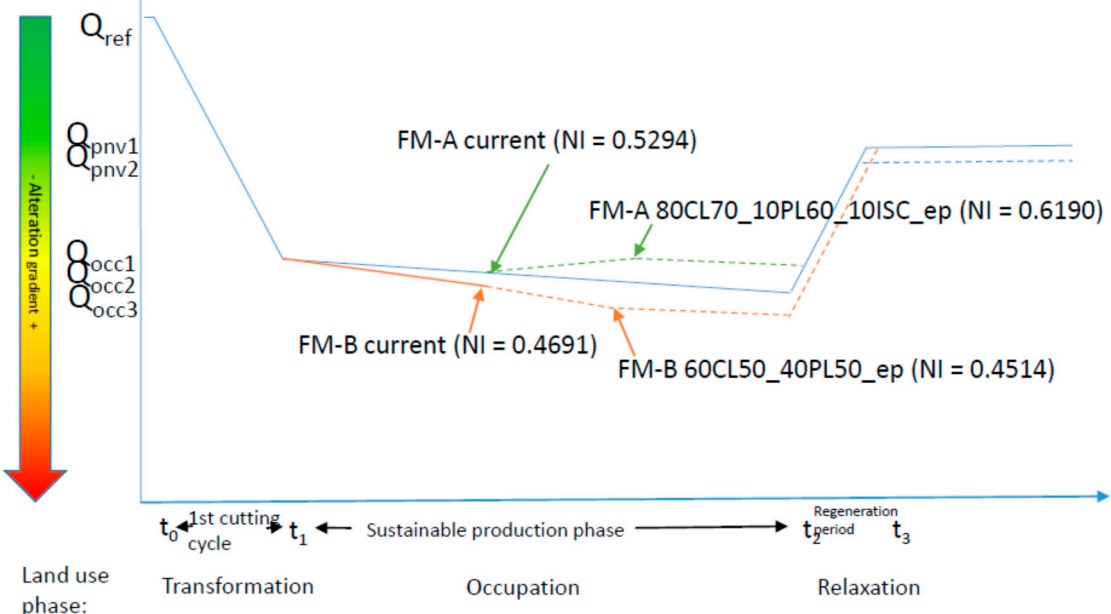

**Figure 5.** Ecosystem quality evolution thru time. $Q_{ref}$: Quality at the reference state (natural); $Q_{PNV}$: Quality of the Potential Natural Vegetation that have been regenerated after land use; $Q_{PNV1}$: with low permanent impacts; $Q_{PNV2}$: with higher permanent impact; $Q_{occ}$: Quality during occupation phase; $Q_{occ1}$: for the scenario with the lower impact; $Q_{occ2}$: for the mid-impact scenario; $Q_{occ3}$: for the lower impact scenario; FM-A 80CL70_10PL60_10ISC_ep: scenario for FM-A figuring 80% of the productive area in clearcut careful logging on a rotation of 70 years, plus 10% in plantation on a 60 years rotation and 10% subject to irregular shelterwood cutting, with the enhanced level of protection; FM-B 60CL50_40PL50_ep: scenario for FM-B figuring 60% of the productive area in clearcut careful logging on a rotation of 50 years, and 40% in plantation on a 50 years rotation, with the enhanced level of protection; NI: Naturalness index.

According to our results, most forest regimes involving a combination of practices (excluding the use of exotic species) applied on a sustainable basis, combined with a significant level of protection (around 30%), would lead to the semi-natural class. However, in FM-B, with a lower naturalness index (NI = 0.4713) compared with FM-A (NI = 0.5294), the threatened boreal caribou has declining populations, though the decline causes are manifold, and result from interacting factors [38,39]. However, in FM-B, where precommercial thinnings were performed over 23.4% of the forested area, the habitat quality of the snowshoe hare, a keystone species in this boreal forest, has been affected; however, effects on populations dynamic depend on multiple factors and are subject to time lag [40].

Our results underline the need for the development of more comprehensive biodiversity indicators that can capture other effects than species loss. The impact on biodiversity of different forest regimes including an important proportion of protection (30%) is related to the impact on species assemblage and populations, rather than to species loss, with the latter being restricted to very specialized and sensitive species. In this respect, a study based on time series showed that changes in assemblage composition has been widespread over the last 40 years, but there was no impact on the number of species [41]. In the *Abies balsamea–Betula papyrifera* forest of low and high altitude, the assemblage of bird species evolves with the vegetation succession [36]. In FM-B, old growth and senescent forests, which are waning, harbor specialized communities of invascular species [37]. In fact, biodiversity issues recognized for the area are mainly related to species associated with old growth and senescent forests or some of their particular features (e.g., dead wood) [38]. This situation raises the issue of

the appropriate biodiversity indicators in LCA, for land uses have a low impact where biodiversity erosion is slow, and do not necessarily result in species loss, in comparison with other land uses causing a drastic change of habitat and observable loss of species richness. The model could thus be used to refine the ecosystem quality assessment in a more comprehensive way than the use of the five naturalness classes.

A test on the scenarios confirms the model's capacity of performing bi-directional assessment to satisfy the need of assessing restoration efforts. However, the effects on biodiversity of enhanced forest management strategies and restoration depend not only on the restored presence of essential habitat features, but also the duration of prior intensive management application [12]. This poses another challenge associated with the use of biodiversity as an indicator of ecosystem quality, as proposed for LCA [9].

Despite over 20 years of research on how to include man-made impacts on biodiversity in LCA, no comprehensive biodiversity impact assessment has been performed so far [42]. In LCA, pressures on biodiversity resulting from land use can be represented as a midpoint impact category, whereas biodiversity in general corresponds to an endpoint category related to ecosystem health [42]. The naturalness assessment model evaluated in this research work could be used as a midpoint level indicator to evaluate the pressure of forestry management practices on ecosystem quality. However, linking such a naturalness index with the endpoint biodiversity indicator, representing the potential loss of species as currently used in LCA, is reductive, as it fails to capture other factors influencing ecosystem health, such as species assemblages and populations. Ideally, other earlier warning biodiversity indicators, such as populations of sensitive species, should be taken into account.

### 4.4. Recommandations for Model's Improvement

As highlighted by the present exercise, here are some recommendations for further use of the model for naturalness assessment:

Historical data should be accurate to avoid bias;

Historical data should include an estimation of the natural variability to improve the setting of the pni's evaluation curves;

Historical data acquisition must match the method used for current evaluation. In the case of methodology improvements—as seen with internal structure evaluation subject to recent technical evolution, or with the new method developed in Quebec's forest inventory for species group identification by 10% of basal area, which would have been appropriate to evaluate the current importance of the white spruce in the stands—the historical data should be accordingly reassessed if possible;

Improve NDP factors evaluation as these are used to evaluate model's most sensitive variables;

Improve the hypotheses used for projection of scenario components on a sustainable basis, considering that the model is applied over the whole landscape (not only on the most important ecological types), as some marginal types in terms of area, such as wetlands, could be important for biodiversity;

Improve the hypotheses used for natural evolution considering that some condition indicators will probably not recover their initial status as a result of past management practices (e.g., some protected area are created in areas which have been subject to harvest in the past) and climate change;

Explore the inclusion of variables characterizing landscape configuration and connectivity [43] in the landscape context.

Tests performed on *Picea* spp. proportion in plantations points to a limit of the model, when using a general indicator, such as the proportion of merchantable volume in spruce, instead of an indicator that would capture the subdominant status of the characteristic late successional companion species.

There is a certain level of uncertainty related to climate change and its potential effects on the natural disturbance regime, which we did not take into account in this study.

Naturalness evaluation should be extended to reach a scale more suitable for LCA. This could be done either by assessing other management units within the same bioclimatic domain [44] (which could also provide a better evaluation of the variability), or be performed at the scale of the bioclimatic sub-domain, depending on the scale of the available data.

Additional research efforts should be dedicated on expanding the assessment on ecosystem alteration levels beyond the sole forestry land use.

## 5. Conclusions

This study has applied a model assessing the impact on ecosystem quality of different forestry management practices through a naturalness index over two regions in Québec. The most sensitive variables for the current naturalness assessment correspond to the regeneration process, the cover type, the dead wood and the closed forest for both territories. The results show the capacity of the naturalness assessment model to perform bi-directional evaluation, assessing not only deterioration, but also improvement of ecosystem quality related to different enhanced ecological management strategies and restoration efforts. In the *Abies balsamea–Betula papyrifera* forest, scenarios that include irregular shelterwood combined with enhanced protection could play an important role in improving ecosystem quality, whereas scenarios applying short rotation (50 years) could lead to further deterioration. Provided that an enhanced protection level is assured, most management scenarios among those tested would produce a semi-natural environment. The model allows adequate forest management scenarios ranking within the different naturalness classes, leading to a finer characterization of the impact of forestry on ecosystem quality. However, the accuracy of historical reference data is important for a fine characterization of the impact, compared with the assessment of the current state of the forest. As most of the results lie in the semi-natural class with an enhanced level of protection, the effects on biodiversity could be mainly related to impacts on species assemblages and populations of species, but not necessarily leading to species losses. This still needs to be better studied.

## 6. Acronyms

| | |
|---|---|
| ANT | Anthropization |
| ANT_NDP | Naturalness degradation potential from anthropization |
| CF | Close forests |
| CF_pni | partial naturalness for close forests |
| CL | careful clearcut logging |
| Compo | composition |
| Compo_PNI | Partial naturalness for composition |
| Context | landscape context |
| Context_PNI | Partial naturalness for landscape context |
| CS | Companion species |
| CS_NDP | Naturalness degradation potential related to companion species |
| CT | cover type |
| CT_pni | partial naturalness index for cover type |
| DW | dead wood |
| DW_NDP | Naturalness degradation potential related to dead wood |
| DW_PNI | Partial naturalness index for dead wood |
| ep | enhanced protection |
| exo | exotic species |
| exo_NDP | Naturalness degradation potential from exotic species |
| for_area | forest or forested area |
| HS | horizontal structure |
| HS_NDP | Naturalness degradation potential related to horizontal structure |
| ip | initial level of protection |
| IR | irregular stands |

| | |
|---|---|
| IR_pni | Partial naturalness index for irregular stands |
| ISC | irregular shelterwood cutting |
| LCA | Life cycle analysis |
| LS | Late successional characteristic species (i.e., *Picea* spp.) |
| LS_pni | Partial naturalness index for late sucessionnal characteristic species |
| NDP | Naturalness degradation potential |
| NE | natural evolution |
| NI | Naturalness Index |
| OF | Old forests |
| OF_pni | Partial naturalness index for old forests |
| PL | Plantation |
| PNI | Partial naturalness index for a given characteristic of naturalness (characteristic_PNI) |
| Pni | Partial naturalness index for a given condition indicator (condition_pni) |
| Prod_area | productive area |
| RP | regeneration process |
| RP_NDP | Naturalness degradation potential related to regeneration process |
| Struc | Structure |
| Struc_PNI | Partial naturalness index for structure |
| W_CC | Clearcuts on wetlands |
| W_CC_NDP | Naturalness degradation potential related to clearcuts on wetlands |
| Wm | Modified wetlands |
| Wm_NDP | Naturalness degradation potential related to modified wetlands |

**Supplementary Materials:** The following is available online at http://www.mdpi.com/1999-4907/11/5/601/s1, Results: Scenario_nb_description: scenario numbering and description; ass_1 and Figure S1: Naturalness assessment for scenarios with enhanced protection (ep) using studies for reference data set; ass_2 and Figure S2: Naturalness assessment for scenarios with enhanced protection (ep) using registry as reference data set; ass_3 and Figure S3: Naturalness assessment for scenarios with initial protection (ip) using studies for reference data set; ass_4 and Figure S4: Naturalness assessment for scenarios with enhanced protection (ep) and PL50 using studies for reference data set; ass_5 and Figure S5: Naturalness assessment for scenarios with initial protection (ip) and PL50 using studies for reference data set; ass_6 and Figure S6: Naturalness assessment for scenarios with enhanced protection (ep) and PL50 with 90% of spruce at maturity using studies for reference data set; ass_7 and Figure S7: Naturalness assessment for scenarios with enhanced protection (ep) and alternate set of NDP factors using studies for reference data set; sensitivity_analysis: results of sensitivity analysis for scenario involving 100% of each practice tested.

**Author Contributions:** Conceptualization, S.C. and L.B.; methodology, S.C. and L.B.; validation, S.C.; formal analysis, S.C.; investigation, S.C.; resources, R.B.; data curation, S.C. and É.T. for dead wood.; writing—original draft preparation, S.C.; writing—review and editing, S.C., L.B., M.M., R.B. and É.T.; visualization, S.C.; supervision, R.B., L.B. and M.M.; project administration, R.B. and M.M.; funding acquisition, M.M. and E.T. All authors have read and agreed to the published version of the manuscript.

**Funding:** This research was funded by the Natural Sciences and Engineering Research Council of Canada (NSERC) through a grant to Manuele Margni, grant number CRD-462197-13, with the collaboration of Cecobois, Canadian Wood Council, Desjardins, GIGA, Hydro-Québec and Pomerleau. The first author received scholarships from: the Discovery Grant to E. Thiffault from NSERC (RGPIN-2018-05755); as well as from the Jean-Claude-and-Lisette-Mercier Fund.

**Acknowledgments:** The authors would like to thank Julie Bouliane from Montmorency Forest for providing data and information, Talagbé Gabin Akpo from Laval University, Mathematics and statistics department consulting service for his help with the data analysis, Stefano Biondo and his team form Laval University GeoStat Center for their help with maps data handling and Naïm Perrault for the localization map.

**Conflicts of Interest:** The authors declare no conflict of interest. The funders had no role in the design of the study; in the collection, analyses, or interpretation of data; in the writing of the manuscript, or in the decision to publish the results.

**Appendix A. Model Adaptation to *Abies balsamea–Betula papyrifera* Domain**

*Appendix A.1. Test Area Description and Localization*

The territory of the Montmorency Experimental Forest, located north of Quebec City and included in the *Abies balsamea–Betula papyrifera* domain, was used as a test area (Figure A1). Most of the current territory of the Experimental Forest was part of a forest concession allocated to the Anglo Canadian Pulp and Paper company in 1926. The territory has been subject to a first commercial harvest in the 1930s and 1940s. At the time of its creation in 1963, the experimental forest covered a total area of 66 km$^2$, now designated as FM-A. In 2014, an adjacent territory of 348 km$^2$, formally part of a public forest management unit with a forest company, has been added to the Experimental Research station and is now designated as FM-B. Therefore, between the 1960s and the beginning of 2010s, the two territories were subject to two different management scenarios. In FM-A, small-scale commercial harvest was continuous since the mid-1960s, while in FM-B, the territory has been subject to a second wave of harvest between 1985 and 2008. At the time of its creation, FM-A was poorly stocked as a result of the young age of the previously harvested stands. Therefore, the initial forest management plan for FM-A focused on the liquidation of the older stands as the harvest priority. Partial cutting was also performed on around 5.5% of the productive area in FM-A and 1.4% in FM-B. Plantations cover 6.8% and 8.8% of the productive area of FM-A and FM-B, respectively. Precommercial thinning has been performed over 9.2% of the productive area for FM-A compared to 23.4% for FM-B. At the time of its inclusion to the experimental station, most of FM-B had been recently harvested; parts of the remaining old forests were saved for protection.

The territory is subject to regular outbreaks of spruce budworm. Past management activities raised the following ecological issues in both territories: decrease of old forests, especially in FM-A, and overabundance of young forests, particularly in FM-B. Prior to the first harvest, stands were characterized by the presence of several old and large white spruce trees. The decrease of spruce in these stands resulting from the first commercial harvest has been described in an experimental design installed on the territory in the 1950s [45]. The remeasurement of some of these plots in the mid 1980s confirmed the scarcity of spruce among the pre-established regeneration of the second-growth forest; spruce regeneration was found only where adult individuals were present, related to residual small trees left at the time of harvest [23]. It also indicated a high risk of broadleaf invasion after clearcutting on the most fertile sites, as a result of the low abundance of coniferous seedlings combined with their small size [23,27].

*Appendix A.2. Model Adaptation to Balsam Fir-White Birch Domain*

For the *Abies balsamea–Betula papyrifera* domain, the five naturalness characteristics of the model were evaluated using the same indicators and variables used for the *Picea mariana*–feathermoss domain of the Quebec's boreal forest [1], except for composition (Table 1). In this case, the decreasing late successional species (LS) are the spruce species. In *Abies balsamea–Betula papyrifera* domain, black spruce (*Picea mariana*) can be dominant on humid stations, but white spruce is a companion species, and has not been quantified at the stand level in the historical data. The percentage of merchantable volume of spruce species for late successional characteristic species was used to assess its presence. However, this measure does not allow controlling for the companion status of the white spruce. As no other companion trees species presents signs of reduction and exotic species were never used in Montmorency Forest, the corresponding values for naturalness degradation potential from exotic species (exo_NDP) and naturalness degradation potential related to companion species diminution (CS_NDP) are set to 0. We kept these variables in the Forest Composition formula (Table 1), despite their null effect for the time being, making provision for future assessment.

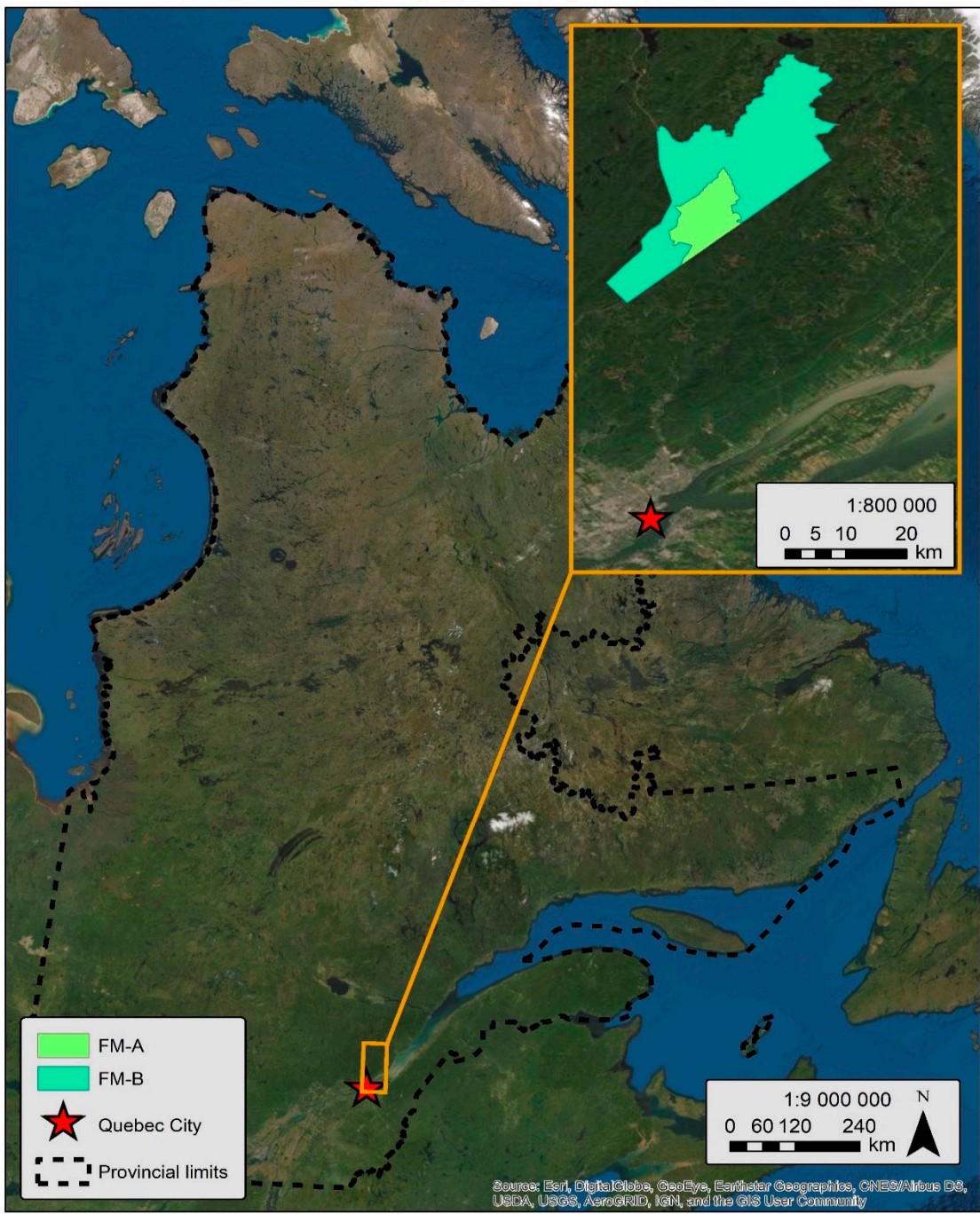

**Figure A1.** Montmorency Forest localization.

*Appendix A.3. Determining Condition Indicator Curves*

Assessing naturalness requires definition of a reference state of the condition indicators, i.e., the natural undisturbed habitat [10]. For Quebec's context, the model application requires pre-industrial data, which can be found in local studies or in the Quebec's reference state registry [21]. We tested both sources of reference data in order to analyze the sensitivity to the historical data set. Model adaptation involves resetting the curves used for partial naturalness index of condition indicators, corresponding to variables related to "pni" in lower cases in the Table 1 equations, using historical values considered valid for the territory. For the original evaluation, reference data used for naturalness assessment were drawn from studies performed in the vicinity of Montmorency Forest [18–20] (lines –s in Table A1).

However, in cases where historical information is not available, studies based on forest dynamic simulation, such as Quebec's reference state registry [21], can be used as an alternate source of historical data. The registry provides reference values for cover types, irregular stands, closed and old forests by homogenous vegetation regions. For the evaluation based on the registry, values for our two sectors were obtained by weighing by the area proportion in each vegetation unit and were used to reset the condition_pni curves accordingly. Compared with the reference data set from studies, the data set from the registry shows, for both territories, less of the coniferous cover type (below current coverage), but a more important coverage of closed forests, old forests and irregular stands (lines –r in Table A1). The spruce proportion was not evaluated in the registry data set; therefore, the same proportion used for the initial assessment was applied for the test using registry's data.

Data used for current naturalness evaluation were taken from the 2018 version of the Quebec eco-forest map [46], except for *Picea* spp. volume proportion [33].

**Table A1.** Historical values from studies (s) in Forest Montmorency vicinity or reference state registry (r) used as reference data, and current values from 2018 eco-forest map.

| Territory | Cover Type (CT: % Forest Area of Coniferous Cover Type) | Late Successional Species (LS: % Merchantable Volume in *Picea* spp.) | Closed Forests (CF: % Terrestrial Area of Forests > 40 Years Old) | Old Forests (OF: % Forest Area of Forests > 80 Years Old) | Irregular Stands (IR: % Forest Area of Irregular Stands) |
|---|---|---|---|---|---|
| FM-A—historical-s | 79.3 [1] | 32 [2] | 79.9 [1] | 23.7 [1] | 17.8 [1] |
| FM-A—historical-r | 63.2 | 32 [2] | 85.5 | 71.0 | 51.1 |
| FM-A—current | 79.1 [3] | 16 [4] | 39.1 [3] | 4.9 [3] | 14.2 [3] |
| FM-B—historical-s | 85.7 [5] | 39 [2] | 76.19 [1] | 57.9 [6] | 40 [7] |
| FM-B—historical-r | 77.8 | 39 [2] | 90.5 | 81.1 | 64.4 |
| FM-B—current | 81.9 [3] | 16 | 27.4 [3] | 16.1 [3] | 18.3 [3] |

FM-A: Montmorency Forest sector a; FM-B: Montmorency Forest sector b; [1] [18]; [2] [20]; [3] Eco-forest map; [4] [33]; [5] Anglo's data in [18]; [6] [19]; [7] Donnacona's data in [18].

Curves used for the evaluation of partial naturalness index of condition indicators evaluation using local studies for reference data are presented in Figure A2 for FM-A and Figure A3 for FM-B. The curves were topped to one for irregular stands (IR), old forests (OF) and coniferous cover type (CT) for the following reasons. Descending curve past the historical value has not been used for IR, because of the evolving evaluation of stands with irregular structure related to improvements of technological identification capacities. For OF, the natural variability is important, as a result of spruce budworm epidemics, and the proportion used as the historical value is based on aerial photography taken around 20 years after an epidemic [18]; the ratio used as the historical value is therefore conservative and values over that ratio are considered natural. The cover type in the Forest Montmorency area was mainly coniferous, and the presence of deciduous cover was related to previous fires and showed an important variability [18]. As the proportion of coniferous cover observed in the ancient forest could reach 97.8% [18], the curve used for pni_CT evaluation was also topped to one. This could theoretically hinder a diagnosis of broadleaf species decrease; however, as no such issue has been identified for that area and no hypothesis used for scenario evaluation goes beyond this proportion, an adjustment for the curve end was not considered necessary. Nevertheless, this situation indicates that it could be appropriate to set both ends of the natural class, considering the natural variability range instead of using a proportion of a unique value. For late successional species (LS), we used the percentage of merchantable volume of spruce species, and descending the curve past the historical value was applied (Figures A2b and A3b), as too much spruce in the landscape would be outside of the natural range of variability.

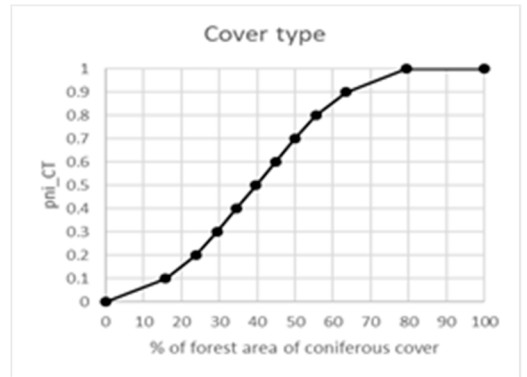

Historical proportion: 79.3% of coniferous cover
Current proportion: 79.1%
CT_pni = 0.998
(**a**)

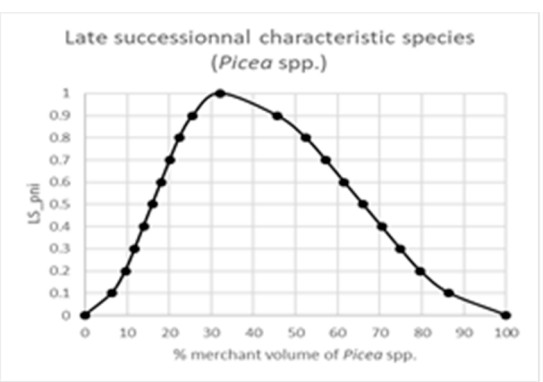

Historical proportion: 32% of *Picea* spp.
Current proportion: 16%
LS_pni = 0.500
(**b**)

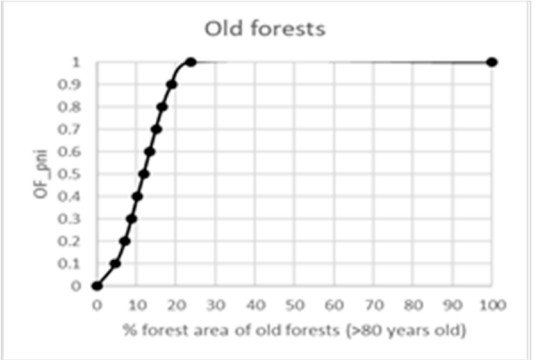

Historical proportion: 23.7% of old forests
Current proportion: 4.9%
CF_pni = 0.106
(**c**)

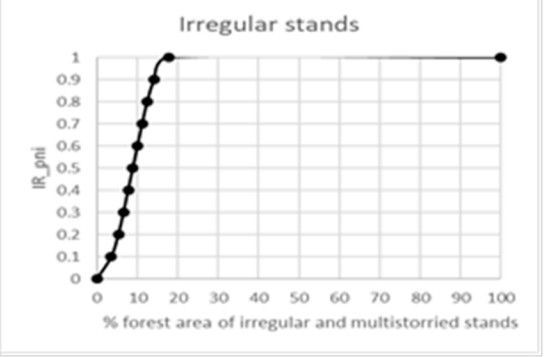

Historical proportion: 17.8% of irregular stands
Current proportion: 14.2%
IR_pni = 0.897
(**d**)

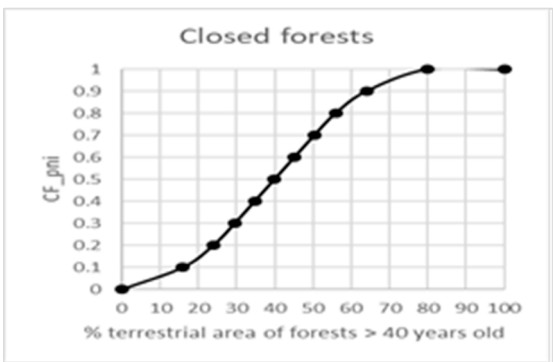

Historical proportion: 79.9% of closed forests
Current proportion: 39.1%
CF_pni = 0.484
(**e**)

**Figure A2.** Curves determining the potential naturalness index to evaluate the condition indicators (condition_pni) for FM-A using local studies for reference data: (**a**) coniferous cover type; (**b**) late successional characteristic species; (**c**) old forests; (**d**) irregular stands; (**e**) closed forests.

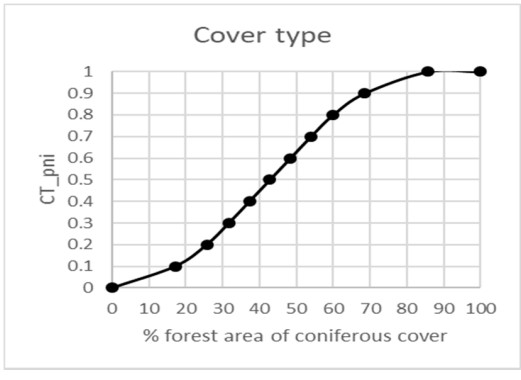

Historical proportion: 85.7% of coniferous
cover
Current proportion: 81.9%
CT_pni = 1
(**a**)

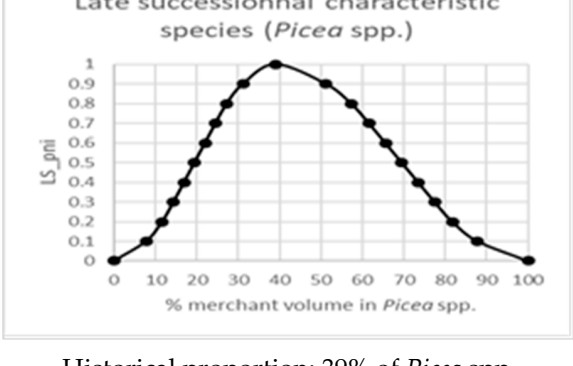

Historical proportion: 39% of *Picea* spp.
Current proportion: 16%
LS_pni = 0.362
(**b**)

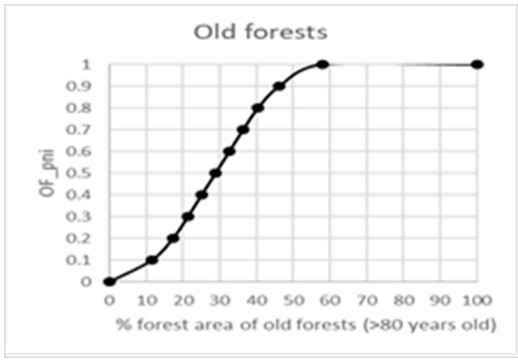

Historical proportion: 57.9% of old forests
Current proportion: 16.1%
CF_pni = 0.178
(**c**)

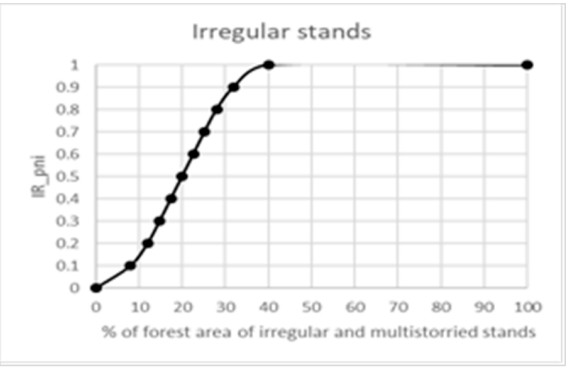

Historical proportion: 40.0% of irregular stands
Current proportion: 18.3%
IR_pni = 0.435
(**d**)

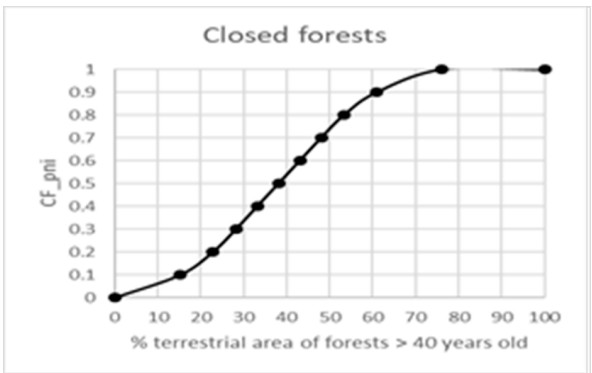

Historical proportion: 76.2% of closed forests
Current proportion: 27.4%
CF_pni = 0.285
(**e**)

**Figure A3.** Curves determining the potential naturalness index to evaluate the condition indicators (condition_pni) for FM-B using local studies for reference data: (**a**) coniferous cover type; (**b**) late successional characteristic species; (**c**) old forests; (**d**) irregular stands; (**e**) closed forests.

*Appendix A.4. Determining the Naturalness Degradation Potentials*

The evaluation of naturalness degradation potentials (NDP) was adapted by resetting the NDP factors related to practices. NDP calculation for horizontal structure, dead wood and regeneration process are presented in Tables A2–A4, respectively, for each territory. For careful logging effects, a distinction has been made between a CL performed in a 50-year-old stand (CL50) and a 70-year-old stand (CL70). Stands resulting from logging performed in 50-year-old stands were identified using an overlay of the eco-forest map and the SIFORT1 map, a tessellation of provincial forest inventory maps for the first measure, considering clear-cuts or careful logging performed between 1981 and 2002, located in sectors having a development stage designed as "young" at the time of the first forest inventory.

The others NDP factors evaluated based on current values and kept constant for all scenarios are showed in Table A5.

**Table A2.** Naturalness degradation potential for horizontal structure (HS_NDP) by silvicultural treatment in Montmorency Forest.

| Territory: | | FM-A | | FM-B | |
|---|---|---|---|---|---|
| Practice | NDP Factors | % Forest_Area | NDPx | % Forest_Area | NDPx |
| Plantation − thinning | 1 | 2.50% | 0.0250 | 5.58% | 0.0558 |
| Plantation | 0.9 | 4.32% | 0.0388 | 3.21% | 0.0289 |
| Thinning (natural), strip cutting | 0.8 | 1.45% | 0.0116 | 0.17% | 0.0014 |
| Precom. thinning (natural), release | 0.75 | 9.24% | 0.0693 | 23.40% | 0.1755 |
| CL50 | 0.5 | 7.01% | 0.0351 | 13.28% | 0.0664 |
| CL70 | 0.3 | 48.34% | 0.1451 | 13.17% | 0.0395 |
| Partial cutting | 0.2 | 5.48% | 0.0110 | 1.38% | 0.0028 |
| Undisturbed or natural disturbances | 0 | 21.64% | 0.0000 | 39.80% | 0.0000 |
| **Current HS_NDP** | | | **0.3359** | | **0.3702** |

Note: NDP_factors: naturalness degradation potential factors related to practices; %for_area: percentage of forested area; NDPx: portion of the naturalness degradation potential for the xth practice; CL50: careful logging (CL) and clearcut of 50-year-old stands; CL70: careful logging (CL) and clearcut of 70-year-old stands.

**Table A3.** Naturalness degradation potential for dead wood (DW_NDP) by silvicultural treatment in Montmorency Forest.

| Territory: | | FM-A | | FM-B | |
|---|---|---|---|---|---|
| Practice | NDP Factors | % Forest_Area | NDPx | % Forest_Area | NDPx |
| Biomass harvesting | 1 | 0.00% | 0.0000 | 0 | 0.0000 |
| Plantation + thinnings | 0.95 | 2.50% | 0.0237 | 5.58% | 0.0530 |
| Plantation − no thinnings | 0.85 | 5.76% | 0.0490 | 3.39% | 0.0288 |
| Partial cutting and precom. thinnings | 0.8 | 14.72% | 0.1178 | 24.78% | 0.1983 |
| CL50 | 0.7 | 7.01% | 0.0491 | 13.28% | 0.0930 |
| CL70 | 0.55 | 48.35% | 0.2660 | 13.17% | 0.0725 |
| Undisturbed or natural disturbances | 0 | 21.64% | 0.0000 | 39.80% | 0.0000 |
| **DW_NDP** | | | **0.5056** | | **0.4454** |
| **Current *DW_PNI*** | | | **0.4944** | | **0.5546** |

Note: NDP_factors: naturalness degradation potential factors related to practices; %for_area: percentage of forested area; NDPx: portion of the naturalness degradation potential for the xth practice; CL50: careful logging (CL) and clearcut of 50 years old stands; CL70: careful logging (CL) and clearcut of 70-year-old stands.

**Table A4.** Naturalness degradation potential for regeneration process (RP_NDP) by silvicultural treatment in Montmorency Forest.

| Territory: | | FM-A | | FM-B | |
|---|---|---|---|---|---|
| **Practice** | **NDP Factors** | **% Forest_Area** | **NDPx** | **% Forest_Area** | **NDPx** |
| Exotic plantations, afforestation | 1 | 0.00% | 0.0000 | 0.00% | 0.0000 |
| Plantation | 0.9 | 6.82% | 0.0613 | 8.79% | 0.0791 |
| Seeding | 0.7 | 2.78% | 0.0195 | 0.00% | 0.0000 |
| Precommercial thinning | 0.65 | 9.24% | 0.0601 | 23.40% | 0.1521 |
| In-fill planting | 0.6 | 2.48% | 0.0149 | 0.00% | 0.0000 |
| CL50 | 0.5 | 4.53% | 0.0226 | 13.28% | 0.0664 |
| Commercial thinning (natural) | 0.4 | 1.45% | 0.0058 | 0.17% | 0.0007 |
| CL70 | 0.35 | 45.58% | 0.1595 | 13.17% | 0.0461 |
| Partial cut | 0.2 | 5.48% | 0.0110 | 1.38% | 0.0028 |
| Undisturbed or natural disturbances | 0 | 21.64% | 0.0000 | 39.80% | 0.0000 |
| **RP_NDP** | | | **0.3547** | | **0.3472** |
| **Current *RP_PNI*** | | | **0.6453** | | **0.6528** |

Note: NDP_factors: naturalness degradation potential factors related to practices; %for_area: percentage of forested area; NDPx: portion of the naturalness degradation potential for the xth practice; CL50: careful logging (CL) and clearcut of 50-year-old stands; CL70: careful logging (CL) and clearcut of 70-year-old stands.

**Table A5.** Naturalness degradation potential for companion species (CS_NDP), exotic species (exo_NDP), wetlands with clear cuts (W_CC_NDP) and anthropization (ANT_NDP) in Montmorency Forest.

| Territory: | FM-A | | FM-B | |
|---|---|---|---|---|
| **Item** | **% Area [1]** | **NDPx** | **% Area [1]** | **NDPx** |
| Companion species | 0.00% | 0.0000 | 0.00% | 0.0000 |
| Exotic species | 0.00% | 0.0000 | 0.00% | 0.0000 |
| Wetlands with clear cuts | 42.37% | 0.2118 | 2.66% | 0.0133 |
| Anthropization | 1.72% | 0.0172 | 2.68% | 0.0268 |

[1] % of forested area for companion species and exotic species; % of terrestrial area for wetlands with clear cuts and anthropization.

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
