# Peer review of "Naturalness Assessment of Forest Management Scenarios in Abies balsamea–Betula papyrifera Forests"

_forests, doi:10.3390/f11050601_

Round 1

Reviewer 1 Report

General comments to authors:

The reviewed article is an interesting example showing application of conceptual model for the assessment of the impact of using wood on the quality of ecosystems. In particular, it is important to use three different scenarios based on two types of reference data. Moreover based on achieved results we can understand witch scenario could lead to further degradation of the ecosystem quality.

Those research are created for Montmorency Experimental Forest, maybe in future it will be possible to create similar analysis in for different case studies.

Detailed comments to authors:

Abstract: Maybe it worth to mention that sensitivity analysis, as a reference point, was created based on two different reference data set: (i) old management plans around the territory of Montmorency Forest, (ii) reference state Quebec’s registry

Line 74: Maybe it is worth to check also different references e.g Cardellini et al. “EFO-LCI: A New Life Cycle Inventory Database of Forestry Operations in Europe”.

Line 155: it is worth justifying the choice of scenarios

Line 352: important difference - whether statistical analysis was performed?

Reviewer 2 Report

The paper presents an application of a model developed in a study of the same authors already published in the same journal (Côté et al, 2019, Forests 2019, 10, 325) and therefore lacks originality. Compared to the previous work, this paper focuses more on the comparison of different management options. The paper is well structured and written, but difficult to follow for readers that do not know the previous paper, where the method is exhaustively described and details on the evaluation steps are given. Reading difficulties are amplified by the use of many acronyms. Other weaknesses are the lack of time references for naturalness and for model predictions.

 Major comments

In general: as many acronyms are used in the text and appendix, a list of them with their meaning, as a separated table, would greatly help the reading. Additionally, some acronyms of the same indicators are different (compare Table 1 and Table A5, Table 1 and Figure 1)

Materials and methods

The model used in the study is not described at all in this section, but the main reference is provided (a previous study of the same authors) and in Appendix A details on adaptations to the new case study are provided. This approach makes the reading difficult for readers who do not know the previous study, and it makes very difficult to understand the introduced changes. I suggest adding few sentences at the beginning of this section on the basic concepts of the model used and especially on the differences (just the indicator composition has been differently evaluated?) with the model developed in the first study.

Reference data for naturalness: the assessment of a degree of naturalness requires a reference “natural” situation, on which authors provide three references from the literature and one from administrative sources. What periods do these sources refer to? In many areas with a long silvicultural tradition (as Europe) it is very difficult or even impossible to establish which is the natural reference vegetation. Is it the potential vegetation? Which and where are the reference examples of the natural conditions for the vegetation types analyzed in this study? To which past time do they refer? The authors address this aspect in section 3.3.2 but they do not provide a reference date or a description on the characteristics of the reference vegetation. Please add some information and/or a discussion point on these issues.

Discussion

The topic of the reference time of model predictions is not discussed: it is not clear at what period the model’s prediction and indicators/variables’ evaluations refer. Scenarios are evaluated without considering that the effects of the silvicultural practices can change over time. A discussion point on this issue should be added.

 Minor comments

p. 1, l. 29: capital letters missing for “natural assessment model"

p. 2, l. 80: the term “bi-directional capacity” is introduced without introducing the concept (ability of the model to assess either the decrease or the increase of naturalness due to wood harvesting) before. In my opinion this possibility should be intrinsic of any model of his type, why do the authors emphasize it?

p. 3, l. 98-100: add the scientific names of the species

p. 7, l. 242-243: here the classification system for naturalness is cited for the first time (semi-natural class 0.4-0.6); please provide the reference of this system, also the list of all classes and related scores among brackets would be useful)

Figure A1: in the legend, for consistency with the text replace FM-A and FM-B with FMa and FMb
